# Endotypes of Prematurity and Phenotypes of Bronchopulmonary Dysplasia: Toward Personalized Neonatology

**DOI:** 10.3390/jpm12050687

**Published:** 2022-04-26

**Authors:** Maria Pierro, Karen Van Mechelen, Elke van Westering-Kroon, Eduardo Villamor-Martínez, Eduardo Villamor

**Affiliations:** 1Neonatal and Paediatric Intensive Care Unit, M. Bufalini Hospital, AUSL Romagna, 47521 Cesena, Italy; maria.pierro93@gmail.com; 2Department of Pediatrics, School for Oncology and Reproduction (GROW), Maastricht University Medical Center, 6202 AZ Maastricht, The Netherlands; karen.van.mechelen@mumc.nl (K.V.M.); elke.kroon@mumc.nl (E.v.W.-K.); 3Statistics Netherlands, 6401 CZ Heerlen, The Netherlands; e.villamorm@gmail.com

**Keywords:** endotype, preterm birth, phenotype, bronchopulmonary dysplasia

## Abstract

Bronchopulmonary dysplasia (BPD), the chronic lung disease of prematurity, is increasingly recognized as the consequence of a pathological reparative response of the developing lung to both antenatal and postnatal injury. According to this view, the pathogenesis of BPD is multifactorial and heterogeneous with different patterns of antenatal stress (endotypes) that combine with varying postnatal insults and might distinctively damage the development of airways, lung parenchyma, interstitium, lymphatic system, and pulmonary vasculature. This results in different clinical phenotypes of BPD. There is no clear consensus on which are the endotypes of prematurity but the combination of clinical information with placental and bacteriological data enables the identification of two main pathways leading to birth before 32 weeks of gestation: (1) infection/inflammation and (2) dysfunctional placentation. Regarding BPD phenotypes, the following have been proposed: parenchymal, peripheral airway, central airway, interstitial, congestive, vascular, and mixed phenotype. In line with the approach of personalized medicine, endotyping prematurity and phenotyping BPD will facilitate the design of more targeted therapeutic and prognostic approaches.

## 1. Introduction

Bronchopulmonary dysplasia (BPD), the chronic lung disease of prematurity, is one of the most common complications of very and extremely preterm births [1,2,3,4,5]. Up to 15–25% of infants below 32 weeks and 60% of infants below 28 weeks develop BPD. BPD is associated with high mortality rates and carries lifelong respiratory and neurodevelopmental consequences [6,7]. BPD is also burdened by high healthcare and social costs and stress for families [1,2,3,4,5,8].

BPD is increasingly recognized as the consequence of a pathological reparative response of the developing lung to both antenatal and postnatal injury [3,4]. Preclinical, clinical, and epidemiologic studies strongly support the contribution of antenatal factors to the pathogenesis of BPD. These factors would act independently or in combination with postnatal insults, such as hyperoxia, ventilator-induced lung injury, or infection [3,4]. According to this view, the pathogenesis of BPD is multifactorial and heterogeneous with different patterns—or endotypes—of antenatal stress that combine with varying postnatal insults and might distinctively damage the development of airways, lung parenchyma, interstitium, lymphatic system, and pulmonary vasculature. This results in different clinical phenotypes of BPD [3,4,5,9]. In line with the approach of personalized medicine, endotyping prematurity and phenotyping BPD will facilitate the design of more targeted therapeutic and prognostic approaches [3,4]. The objective of this review is to summarize and discuss the current knowledge and future perspectives on the association between endotypes of prematurity and phenotypes of BPD.

## 2. Historical Perspectives of BPD Pathogenesis and Diagnostic Criteria

The evolution of perinatal care over the past few decades has affected premature birth and its complications, including BPD. When BPD was first described by Northway in 1967 [10], the main histopathological hallmark was a high degree of fibrosis and inflammation as a consequence of the aggressive ventilatory strategies on a relatively mature lung in the saccular stage of lung development (32–36 weeks’ gestation) (Figure 1). Northway described four stages of lung disease, according to chest radiographic findings and changes in histopathology, going from the acute to the chronic phase of the disease that developed after 28 days of life.

Advances of prenatal and neonatal care, such as antenatal corticosteroids [11], surfactant replacement therapy [12], or gentler modes of ventilation [13] not only reduced the impact of ventilator-induced lung injury (VILI), but also enabled the survival of infants born at earlier stages of development. Oxygen dependency at 28 days of life was found not to be a reliable indicator of BPD for the new preterm population needing intensive care. This evidence led to the revision of the BPD definition as oxygen dependency at 36 weeks post menstrual age (PMA) (Figure 2) [14]. At this time, the terminology of chronic lung disease was also introduced as an alternative to BPD. In 1999, Jobe coined the term ‘new BPD’ to describe the characteristics of a condition mainly characterized by the arrest of lung development at the canalicular stage (Figure 1) [15]. This insight motivated another refinement of the definition of BPD, in the workshop held by the National Institute of Child Health and Human Development (NICHD) in 2001 (Figure 2) [16]. The 2001 NICHD definition required two criteria in order to diagnose BPD: the weeks of gestation below 32 and the need for supplemental oxygen beyond 28 days of life and/or oxygen dependence at 36 weeks PMA [16]. The 2001 NICHD definition also categorized BPD into mild, moderate and severe based on the required amount of oxygen and/or ventilatory support at 36 weeks PMA. A further reconsideration of the definition was the inclusion of a physiologic challenge named “room air challenge test”. Based on this physiologic definition, infants are classified as having BPD at 36 weeks PMA if their oxygen saturation drops below 90% following the withdrawal of supplemental oxygen for one hour [17].

During the past decade, less invasive forms of non-invasive respiratory support have been implemented for preterm infants [13]. These devices have no place in the BPD definition (Figure 2). Moreover, infants who die from the disease before 36 weeks PMA, due to the most severe form of lethal BPD, were not considered. To address these gaps, the definition of BPD was again modified in 2016 [18]. The 2016 NICHD definition includes infants born at less than 32 weeks’ gestation, having radiographic-confirmed lung disease and requiring ventilator support for more than three consecutive days, either invasive or non-invasive, to maintain arterial oxygen saturation above 90% at 36 weeks PMA [18]. The disease severity nomenclature was modified into three grades. Grade III would be the most severe form of BPD. Grade III(A) is a new category that includes infants who die between 14 days of postnatal age and PMA of 36 weeks because of parenchymal lung disease and respiratory failure [18].

## 3. Current Definitions of BPD: Downsides and Pitfalls

Although trying to take into consideration the changing times, all the BPD definitions that have been coined over the years are hampered by several pitfalls. The major limitation is the fact that all definitions are based on the need for a therapy (i.e., supplemental oxygen) instead of on etiopathogenesis. Moreover, oxygen therapy is highly inconsistent among institutions and even clinicians. The lack of standardization of saturation targets, especially in ex-preterm infants approaching corrected term [19,20], results in discrepant BPD rates. The saturation threshold of 90% may be too low as normal saturation levels in healthy term infants should be above 96% [21]. Therefore, preterm infants suffering from lung disease leading to mild impairment in oxygenation may not be recognized as BPD patients when using these criteria. Moreover, oxygen targeting is particularly challenging in infants with evolving or established chronic lung disease. This is mainly due to a combination of apneas and significant ventilation-perfusion disturbances in the premature lung causing unpredictable phases of intermittent hypoxemia and hyperoxia [22]. Therefore, preterm infants requiring supplemental oxygen in the NICU (Neonatal Intensive Care Unit) usually spend half of their time outside the prescribed target range [23]. The time spent in the appropriate range depends on several variables, including the chosen target of oxygen saturation and the clinician and nurse levels of tolerance to hypoxemia and hyperoxia phases [24,25]. In addition, oxygen requirements do not only reflect lung disease but other factors such as altitude, presence of comorbidities and medications [26,27]. These discrepancies further influence the data on BPD rates.

Besides the poor ability in providing reliable rates of the disease, oxygen need at 36 weeks PMA may not predict long-term outcomes. The need for ventilatory support has been proposed as a better way to correctly foresee death or severe respiratory morbidity regardless of the supplemental oxygen use [28]. Among infants needing respiratory support at 36 weeks PMA, a BPD severity classification based on the mode of respiratory support (invasive versus non-invasive) further improves outcome prediction [29].

Aside from the type of support chosen to define BPD and its severity, the time point at diagnosis is another unsolved issue. Isayama et al. explored different time-points to diagnose BPD and predict respiratory outcomes [30]. The need for oxygen at 40 weeks PMA was identified as the best predictor for serious respiratory morbidity. Finally, the major shortcoming of all BPD definitions is the lack of pathophysiological clues that may address targeted treatment in order to improve quality of life and slow the progression of the disease. While the duration and the different levels of respiratory support may be appropriate to score disease severity, those criteria do not allow guiding a dynamic diagnostic process and the resulting therapeutic approach.

## 4. Endotypes of Very and Extremely Preterm Birth: Definition and Impact on Fetal Lung Development

The term endotype refers to “a subtype of a condition, which is defined by a distinct functional or pathophysiological mechanism.” [31]. Endotypes are thus a different form of classification from clinical phenotypes and describe distinct disease entities with a defining etiology and/or a consistent pathophysiological mechanism [31,32].

There is no clear consensus on which are the endotypes of prematurity but the combination of clinical information with placental and bacteriological data enables the identification of two main pathways leading to birth before 32 weeks of gestation: (1) infection/inflammation and (2) dysfunctional placentation. The first group include chorioamnionitis, pre-labor premature rupture of membranes, placental abruption, and cervical insufficiency [33]. The second group is characterized by a relative absence of microorganisms and inflammation, but the presence of histologic features of placental vascular dysfunction is associated with hypertensive disorders of pregnancy (HDP) and the entity identified as fetal indication/intrauterine growth restriction (IUGR) [33].

The environment to which the developing lung is exposed varies greatly between the two endotypes of prematurity. The possible mechanisms by which infection and dysfunctional placentation can interfere with lung development have been extensively explored in preclinical studies [34,35,36]. Antenatal exposure to endotoxin to mimic chorioamnionitis induced sustained abnormalities of distal lung growth and pulmonary hypertension in murine and ovine models [34,37,38,39,40,41]. This effect appears to be mediated by changes in the expression and/or activity of various cytokines and mediators acting as regulators of alveolar development, angiogenesis, and airway remodeling [34,36,37,38,39,40,41]. The biological mediators most frequently suggested to be involved are transforming growth factor (TGF)-Beta 1, interferon-γ-inducible protein (IP)-10, connective tissue growth factor (CTGF), nitric oxide synthase (NOS), hypoxia inducible factor (HIF), vascular endothelial growth factor (VEGF), and platelet endothelial cell adhesion molecule (PECAM)-1 [34,36,37,38,39,40,41].

Among the prenatal infectious/inflammatory stimuli, the one mediated by *Ureaplasma* spp. has received particular attention. *Ureaplasma* infection of the developing lung has been studied primarily in sheep and non-human primates [42,43,44]. In general, the lung alterations were more severe in the latter animal models, suggesting species differences in susceptibility to *Ureaplasma* [44]. Colonization with *Ureaplasma urealyticum* led to severe bronchiolitis and interstitial pneumonitis in fetal baboons [42]. Interestingly, colonized fetuses that subsequently cleared *Ureaplasma* pulmonary colonization demonstrated early improvement in lung function [42]. Exposure of fetal sheep to *Ureaplasma parvum* increased surfactant production but induced a mild acute inflammatory response that altered lung deposition of elastin and α-smooth muscle actin [43,44]. In addition, preclinical evidence suggests that antenatal exposure to *Ureaplasma* can modulate the innate immune system and prime the immature lung to be more vulnerable to inflammatory stimuli after birth [43,44].

Regarding placental dysfunction, it has been suggested that both the fetal lung and the placenta have similar responses to alterations in the intrauterine environment [35,45]. This may be particularly true for responses to hypoxia [35,45]. Preclinical models of placental dysfunction show profound impairment in alveolar and lung vascular growth, even in the absence of postnatal adverse exposures [35,36,39,45,46]. Growing evidence suggests that the pathogenesis of these alterations involves a complex interplay of pro- and antiangiogenic factors, oxidative stress, epigenetic alterations, and activation of the endoplasmic reticulum stress pathway [35,36,39,45,46].

## 5. Endotypes of Prematurity and Respiratory Outcome

Although it is a common understanding among neonatologists that the pathophysiological condition triggering preterm birth plays a major role in the clinical picture and the development of complications, a persistent conundrum is which part of these complications are due to prematurity and which part are due to the pathological changes induced by the endotype. Comparing how different pathological conditions affect the same outcome can provide insight into the relationship between endotype and phenotype.

Very recently, we conducted two meta-analyses on the association between endotypes of prematurity and BPD [47,48]. In these analyses, the proxy for the infectious/inflammatory endotype was chorioamnionitis [47], while the placental dysfunction endotype was represented by HDP and fetal growth restriction [48]. Since most studies defined growth restriction as small for gestational age (SGA), this group was referred to as SGA/IUGR. As summarized in Figure 3, chorioamnionitis was associated with an increased risk of developing BPD as defined by the need for oxygen at 28 days (any BPD) or at 36 weeks PMA (moderate/severe BPD) [47]. In contrast, meta-analysis could not detect a significant association between HDP and BPD but it did show a significant association between SGA/IUGR and risk of both moderate/severe BPD and severe BPD [48] (Figure 3). A high degree of heterogeneity was detected in all the analyses [47,48] (Figure 3).

Our studies also confirmed that key factors for the development of BPD are distributed differently in the chorioamnionitis, HDP, and SGA/IUGR groups [47,48]. When compared to their respective control groups, infants exposed to chorioamnionitis showed a significantly lower GA [47], while those in the HDP and SGA/IUGR groups showed a significantly higher GA [48]. Meta-regression analysis showed that these differences in GA correlated with the effect size of the association between prenatal insults and BPD and were responsible for an important part of the heterogeneity observed in the meta-analyses [47,48]. Interestingly, when the meta-analysis was limited to studies with no difference in GA between the exposed and the unexposed group, the positive association between chorioamnionitis and BPD could no longer be demonstrated [47], but an association between HDP and BPD risk was unmasked [48].

Taken together, the data from these two meta-analyses suggest that chorioamnionitis (i.e., the infectious/inflammatory endotype) has a greater overall impact on the risk of developing BPD as it is the most frequent endotype in the lower and more vulnerable GA [47]. However, when the endotype of placental dysfunction is accompanied by fetal growth restriction, it is strongly associated with higher rates and severity of BPD even though newborns are more mature [48]. Moreover, BPD associated with placental vascular dysfunction may have a greater component of vascular disease manifested as pulmonary hypertension [48] (Figure 3).

In addition to differences in GA, our meta-analyses also showed that chorioamnionitis was associated with higher rates of antenatal corticosteroids [47] and that HDP was associated with higher frequencies of female fetuses [48]. These two factors may modulate the role of the endotype on the development of respiratory complications [49,50,51] and require further investigation. Finally, we also analyzed the association of chorioamnionitis, HDP, and SGA/IUGR with the risk of developing respiratory distress syndrome (RDS). It is noteworthy that both chorioamnionitis and placental dysfunction have been considered as inducers of an early-protection, late-damage effect. This is a reduction in RDS but an increase in BPD [34,35]. Although very limited by the heterogeneity in both the statistical and clinical definitions of RDS, our results suggest that the lower incidence of RDS is observed only in the SGA/IUGR group [47,48] (Figure 3).

## 6. Phenotypes of BPD

BPD is an umbrella term that may encompass different forms of lung injury, caused by various mechanisms and characterized by specific molecular pathways. These mechanisms may affect, in a relatively specific way, airways, alveoli, vessels, interstitiums, and lymphatic systems (Figure 4), giving rise to different clinical phenotypes of BPD [52]. A phenotype describes the observable physical manifestations of a particular disease. The term phenotypes of BPD has been used with disparate meanings, including: phases of the disease, ‘old’ or ‘new’ BPD, disease severity, radiographic classification, lung function, or injury of a specific part of the respiratory system. [52,53,54]. The latter will be considered as BPD phenotypes in this review. This notion of BPD phenotypes implies that the need for oxygen or respiratory support at 36 weeks PMA is a common symptom resulting from a variable involvement of the different pulmonary compartments. Wu et al. performed a retrospective study of a cohort of preterm infants with severe BPD, aiming to define the frequency of three BPD phenotypes [54]. Approximately 12% of the included infants displayed features of pure parenchymal lung disease, 8% had mainly pulmonary hypertension, and 7% had large airway disease as a major cause for their oxygen dependency. The most common phenotypic presentation (32%) was the co-occurrence of all three components. Other combinations of phenotypes were also present. From a prognostic point of view, pulmonary hypertension and large airway disease were related to a worse outcome, including death prior to discharge, tracheostomy, or pulmonary vasodilator use when discharged [54]. Beside the retrospective nature of the analysis, this study has other limitations. Not all the infants were studied for all the three components. Approximately one-third of the included infants did not undergo tracheoscopy and/or bronchoscopy, limiting the accurate assessment of the central airway unit. Moreover, this study included only infants affected by the most severe form of the disease, as confirmed by the fact that almost 60% of the infants were still on mechanical ventilation at 38–40 weeks PMA and 90% were discharged on home oxygen therapy. Since pulmonary hypertension is associated with the most severe forms of BPD [55,56], there is a significant risk of selection bias in this study. The assessment was done only one time, while different time-points may be needed to obtain a full assessment of the phenotypes. In addition, it is likely that more than three phenotypes are present in the BPD spectrum, especially from a diagnostic and therapeutic standpoint. Despite these limitations, this study has the great merit of first unveiling different clinical entities among BPD diagnosed patients.

The approach to BPD classification according to the phenotype hypothesis may carry important clinical consequences. Future BPD classifications, according to the endotypes and phenotypes of the disease, may guide specific diagnostic approaches and consequent plausible treatments. Below is a tentative description of the BPD phenotypes that may require tailored approaches during either the evolving or the established phase of the condition (Figure 5). This description gathers the current knowledge on BPD pathogenesis and long-term functional consequences and borrows evidence from adult pulmonary diseases that share similarities with potential BPD phenotypes.

### 6.1. Parenchymal Phenotype

The parenchymal phenotype of BPD is mainly characterized by the arrest of development in the alveolar portion of the lung, which results in larger and fewer alveoli and reduced alveolar surface area leading to impaired gas exchange [57]. These features resemble the emphysematous form of chronic obstructive pulmonary disease (COPD) [58]. Emphysema has been associated to BPD in terms of mechanisms of lung injury and subsequent molecular cascades [59]. Moreover, the disruption of alveolar growth associated with BPD represents a possible link to early onset COPD in adult life [60]. A parenchymal disorder was found, alone or in combination with other components of respiratory disease, in up to 78% of preterm infants with severe BPD undergoing a computed tomography (CT) scan [54]. The diagnosis of the parenchymal phenotype of BPD was made according to a CT scan-based score [61], which considers the degree of hyperexpansion and emphysema, as well as the fibrous/interstitial abnormalities [61]. The emphysematous features of lung damage are very likely different from the interstitial alterations on a functional, diagnostic, and therapeutic basis. Emphysema is an airflow-limited disease, characterized by increased compliance and obstructive airflow limitation as opposed to the restrictive pattern typical of the interstitial disorders. Approximately 60% of preterm infants suffering from BPD develop an obstructive disease, 10% a restrictive disease, and 30% a mixed form [53]. Chest imaging of children with BPD has demonstrated abnormalities suggestive of airflow obstruction. Chest CT scans can detect areas of hyperexpansion and hyperlucency in children with a history of BPD [62]. However, there are a lack of robust studies correlating lung CT imaging scores with lung function and clinical symptoms in infants with BPD [63,64]. Currently no therapeutic options are available for the emphysematous lung arrest due to BPD. Fortunately, alveolar septation, the process responsible for the increase in gas exchange surface area, takes place from 36 weeks gestation to 3 years of postnatal life (Figure 1). As a consequence of alveolar multiplication, parenchymal disease can improve over time and patients needing home oxygen may be weaned off from respiratory support successfully [65,66]. However, many children and young adults with a history of preterm birth still present with exercise limitation and reduced pulmonary reserve, even with normal lung capacity [67,68,69]. A better definition of the parenchymal phenotype and the striking similarities with emphysema may facilitate the development of common innovative strategies that may be tested in experimental models and clinical studies.

### 6.2. Peripheral Airway Phenotype

The small airways component of BPD, consisting of structural remodeling, bronchoconstriction, and hyperreactivity, manifests similarly to asthma [70]. However, Baraldi et al. reported that the survivors of BPD showed, at school age, low exhaled nitric oxide levels and poor response to β2-agonists during the evaluation for airflow limitation [71]. This is quite different from the conditions of other typical asthmatic children and suggests that the pathogenesis of obstructive lung disease in patients with BPD might involve structural changes in small airways in addition to airway inflammation [71,72]. Similarly to COPD, the obstructive lung disease related to BPD may include a structural fixed component and a reactive inflammatory component. The former may respond poorly to standard asthma therapies while the latter may be more responsive to treatment [70]. In adults suffering from COPD, the Global Obstructive Lung Disease (GOLD) system has been used to identify and classify the severity of post-bronchodilator airflow limitation [73]. It is noteworthy that individuals with identical GOLD stages may have different morphologic appearances at CT [74]. Some have extensive emphysema, whereas others with equal functional impairment have an airway-dominant phenotype with little or no emphysema. These morphologic differences may reflect important differences in the underlying pathophysiology and response to therapy. Interestingly, post-processing CT techniques demonstrated a combination of areas of low and high attenuation in BPD patients combining a small airways component with areas of emphysema [75]. From a lung function point of view, both the parenchymal and the small airway phenotypes could be classified as obstructive conditions.

### 6.3. Central Airway Phenotype

Tracheomalacia, subglottic stenosis, bronchomalacia, and bronchial stenosis are the clinical manifestations of the central airway diseases phenotype of BPD [3]. Central airway collapse depends on the rigidity of the central airway and the collapsing transmural airway pressure [76]. Airway compliance is inversely proportional to the GA of newborns. The immature tracheal cartilage is hypercellular with little glycosaminoglycans. This results in a more compliant, smaller and more prone to collapse airway, compared to term infants [77]. The softness of the immature airway is vulnerable and easier to deform when exposed to positive pressure ventilation [77,78], which further weakens tracheal and bronchial walls and predisposes them to collapse during expiration [78,79,80]. Airways of infants with BPD are characterized by smooth muscle hypertrophy, thickening of airway walls, epithelial inflammation, and septal and parenchymal fibrosis [77,78,80], which can result in increased peripheral airway resistance [81]. This, in addition to the fact that children with obstructive respiratory disease often use accessory muscles to exhale, result in an increase in collapsing transmural airway pressure [77]. Prevalence studies have demonstrated that the incidence of intrathoracic tracheo-bronchomalacia in preterm infants undergoing flexible bronchoscopy may range from 16 to 50% [54,82]. However, this prevalence may be underestimated because of unawareness, lack of diagnostic criteria [83], limited availability of bronchoscopic examination [84], and because the procedure is often performed only in a small number of preterm infants [70,85]. Both cholinergic agonists and antagonists may be used in the clinical management of malacia, with no evidence to support either approach [77]. The most severe cases may need tracheostomy. Intraluminal stenting is not usually performed because of risk of complications, and extraluminal stenting requires major thoracic surgery [86]. Tracheomalacia may improve in the first two years of life.

### 6.4. Interstitial Phenotype

Interstitial disorders, such as an increase in fibrotic tissue and widening of the interstitial spaces, are frequently observed in ex-preterm infants suffering from BPD and undergoing lung biopsy during childhood [57,87]. This suggests that the interstitial phenotype of BPD may be a distinct entity from the parenchymal phenotype [58]. Interstitial injury may be worsened by capillary leak from inflammation due to infection, RDS, or ventilator-induced lung injury. From a lung function point of view, the interstitial phenotype is likely to show restrictive features. However, despite radiological and functional evidence of interstitial injury in BPD [88], there is a lack of diagnostic criteria for this phenotype. Some forms of BPD fulfill the radiological and histopathological criteria included in the classification of children’s interstitial lung diseases (chILD) [75,87,89,90,91]. However, the uniqueness of the interstitial phenotypes, which may start from the evolving phase of the disease, goes largely unnoticed. In adult patients suffering from interstitial lung disease, the use of high doses of steroids is proven to significantly improve survival [92]. Being able to select the patients with the interstitial phenotype may allow targeting patients who could benefit from steroid treatment at an early stage of the disease.

### 6.5. Congestive Phenotype

Infants with BPD are prone to pulmonary edema, presumably due to immature and inadequate pulmonary lymphatic drainage. Pulmonary edema may be exacerbated by the presence of a patent ductus arteriosus with left to right shunt or by inflammation resulting in capillary leak [93]. Excessive interstitial edema leads to decreased lung compliance and impaired gas exchange. The presence of pulmonary edema often results in changes in surfactant activity and increases surface tension, which may result in the development of atelectasis [94]. Lung atelectasis increases the risk of infection, alters gas exchange and increases lung injury. In addition, prematurity is associated with systemic arterial stiffness [95], possibly resulting in left ventricular hypertrophy and dysfunction from excessive afterload, which can further worsen pulmonary edema [93]. Chest x-ray is currently the standard method to detect lung overload, although it is not specific. Although diuretic therapy was not effective in preventing BPD [96], a target therapy in the congestive phenotype of BPD may alter the course the disease and improve the outcome (Figure 5).

### 6.6. Vascular Phenotype

The vascular hallmark of BPD is a dysmorphic capillary bed with dysregulation of pulmonary vascular development characterized by vascular growth arrest, reduced distribution of pulmonary capillaries, and altered pattern of vascular organization [97,98,99]. As mentioned above, the vascular phenotype of BPD appears to be strongly associated with the endotype of placental vascular dysfunction [48] (Figure 3). Postnatal stimuli, such as hemodynamic stress, hyperoxia, and hypoxia, may further impair pulmonary vascular development, leading to smooth muscle proliferation and integration of myofibroblasts and fibroblasts into the vessel walls [99]. These changes result in increased vascular narrowing, decreased vascular compliance and subsequent increased pulmonary vascular resistance. The clinical result of this dysfunctional lung vascularization is pulmonary hypertension that may begin as early as the first few weeks or manifest itself more clearly when BPD is already developed [100,101]. Studies showed a strong association between early and late pulmonary hypertension in BPD patients [101], suggesting that the phenotypes of BPD may be a continuum from an acute to a chronic disease, being possibly influenced by the endotype of prematurity [100].

Retrospective studies show that late pulmonary hypertension may occur in up to 17–43% of infants with BPD [102,103,104] and that the mortality rate of the condition may be as high as 14–38% [98,105]. Consensus diagnostic criteria for BPD-associated pulmonary hypertension are not available [106]. The presence of pulmonary hypertension is usually assessed by a multiparametric echocardiographic approach, evaluating different indirect measurements of arterial pulmonary pressure associated with the interpretation of left and right ventricular function [106]. However, more advanced non-invasive (magnetic resonance imaging, CT) and invasive (cardiac catheterization) techniques may be needed to assess complex cases. These techniques allow a better assessment of heart function and the eventual evaluation of pulmonary vein stenosis, thromboembolic events, lung perfusion impairment, and response to vasodilators [106,107,108,109,110]. Unfortunately, these tools are not available for neonates in most centers.

### 6.7. Mixed Phenotype

During pulmonary development, there is a constant interaction between the different lung compartments. The most conspicuous example is the interaction between the processes of alveologenesis and vasculogenesis [111,112]. In addition, and depending on the timing, nature, and intensity of prenatal insults, there will be a variation in the degree of damage to the various lung compartments. Therefore, the different BPD phenotypes will not develop as single isolated conditions but will most often be a mixed phenotype with varying degrees of involvement of the different pulmonary compartments [53]. These mixed phenotypes might be the most severe ones in terms of mortality and morbidity. Addressing each single component of mixed BPD phenotypes may be needed to develop the appropriate therapeutic strategy.

## 7. Prematurity Endotypes and BPD Phenotypes: Implications for Personalized Clinical Management and Research

### 7.1. Personalized Diagnosis and Follow-Up

Changing the approach from a definition based on signs and symptoms to a causal definition of BPD, based on the chain of events that underlie the disease, would substantially change the way BPD is diagnosed (Figure 5). Moreover, the approach on specific phenotypes and phenotype evolution would reduce the debate on the appropriate timing as well as the most suitable clinical data chosen for the diagnosis. The major concern to this approach is that the currently available tools to reach a more precise phenotype interpretation are either invasive, require exposure to radiation, or are not accessible for most centers.

The field probably demands a composite assessment so that the predictive ability of a definition is reasonable, cost-effective, and easy to use. Easily available biomarkers and bedside testing combined with safe, non-invasive, radiation-free, accessible instrumental tools are needed to account for BPD phenotypes longitudinally from birth into further childhood. Point of care markers such as pro-BNP have been used in the assessment of the progression of pulmonary hypertension, although they may be influenced by other variables such as PDA [106]. The implementation of point of care imaging, such as lung ultrasound in association with functional echocardiography, may help distinguish interstitial from congestive phenotypes [113]. Echocardiography can help diagnosing and assessing vascular phenotypes. Lung function tests may distinguish restrictive from fixed or variable obstructive forms and guide potential treatments [58]. Quantitative and qualitative assessments would optimally apply to the different phases and time-points of BPD phenotypes, and can serve as clinically significant measurements for therapies or interventions during the course. Finally, the possibility to follow over time the different components of BPD may help to better understand the disease progression in those infants who evolve to the more severe, potentially lethal, forms of BPD [18] and who may benefit from specific life-saving therapies.

### 7.2. Toward a Personalized Therapy for BPD

A phenotype-based definition of BPD should take into account the different pathophysiological pathways and may lead to more specific treatments that not only decrease symptoms, but also change the course of the disease (Figure 5). The possibility to timely diagnose the different phenotypes may not only help target treatment, but also reduce exposure to unnecessary therapies and limit side effects. In addition, the use of specific treatments depending on BPD phenotype may increase the therapeutic efficacy of drugs, such as diuretics, whose overall effectiveness in BPD is limited [114]. It may also allow an optimization of the use of corticosteroids that, although effective for BPD, have important side effects [114].

Infants with a predominant small airway BPD phenotype may have variable responses to bronchodilator and anti-inflammatory drugs, depending on the degree of fixed and reactive components of airway obstruction. Infants with a parenchymal emphysematous phenotype, who are unlikely to respond to bronchodilator therapies, as well as infants with a fixed small airways component of BPD may be candidates for future trials investigating novel therapies. The possibility of an early distinction between these phenotypes would allow the early start of anti-inflammatory or bronchodilator therapy (Figure 5). In addition, infants that are unlikely to respond to standard therapies may be on a “fast-track” to novel experimental therapies.

Infants suffering from pulmonary hypertension associated to BPD (i.e., vascular phenotype) often present with recurrent cyanotic episodes, have prolonged initial hospital admissions, and require extra oxygen supplement or higher levels of respiratory support even after discharge compared with infants with isolated BPD [110]. In the long term, children with chronic pulmonary vascular disease will have different degrees of exercise intolerance and persistent echocardiographic abnormalities along with cardiac remodeling [110]. Untreated pulmonary hypertension may lead to right ventricular strain and dysfunction that may result in right ventricular failure. Moreover, the cardiac status may be complicated by the systemic arterial stiffness of very preterm infants which can result in sufficient afterload to create high end-diastolic pressure, left ventricular hypertrophy and dysfunction [57,95]. Treatment may start with diuretics and then move towards specific vasodilators [106] (Figure 5). Even among preterm infants of similar gestational ages, extreme variability in vascular disease exists and reflects the variable responses to vasodilators.

### 7.3. Research

Classification and definition of prematurity endotypes and BPD phenotypes may improve the design of clinical and translational studies [3]. This would enable early identification of at-risk preterm infants and individualized care to enhance outcomes [3]. Since it is very unlikely that a single medication can be effective to treat every BPD phenotype, clinical trials should try to provide a tentative description of the phenotypes of the included patients in order to run subgroup analysis or logistic regressions. In addition, large multicenter genomic studies that include multiple ethnicities, antenatal risk factors and careful definition of phenotypes, may provide important clues about pathogenetic pathways and BPD phenotypes [3]. Unfortunately, the growing interest in BPD phenotypes is still scarcely translated into the design of clinical studies. When in April 2022 we carried out a search of the RCTs registered in www.clinicaltrials.gov (accessed on 11 April 2022), we found that only four of the 91 ongoing studies referred to a BPD phenotype. This was always the vascular phenotype represented by BPD-associated pulmonary hypertension. Similarly, of the approximately 80 studies on BPD completed in the last 10 years, only five took into account a possible phenotype, which was again the vascular phenotype.

In addition, the connection between endotypes of prematurity and phenotypes of BPD should also be investigated in the preclinical setting. The most frequently used animal models of BPD are based on the hyperoxic insult [115,116]. These models do not reproduce the complete pathophysiology of BPD or the interaction between endotype and phenotype. This may explain, at least partially, the discrepancies between preclinical and clinical studies observed in the effects of therapies such as inhaled nitric oxide [114,117]. Interestingly, research is now focusing on models of BPD that take into account the endotypes of prematurity, such as the combination of chorioamnionitis with hyperoxia [41] or preclinical models of preeclampsia-induced lung injury [118].

### 7.4. New Challenges: BPD and COVID-19 Pandemic

COVID-19 is generally a mild disease in children. However, a proportion of infants that test positive for SARS-CoV-2 require admission to the hospital [119]. Among hospitalized children aged <2 years, chronic lung diseases, including BPD, were the most important risk factor for the development of severe COVID-19 [120].

Besides the effect of the viral infection on patients with BPD, the changes in lifestyle pose these infants at high risk for a worsening of their condition and quality of life. The fear of contagion can influence infants’ life in many ways. Patients with chronic lung disease will be at particular risk considering that they may experience stigmas related to their respiratory symptoms such as chronic cough and being at risk of isolation by their peers [121]. Moreover, 30.7% of the subjects have skipped their regular follow-up visits due to the fear of catching the virus in healthcare facilities. In addition, lifestyle changes due to the pandemic may have detrimental effects, especially in children with chronic lung disease, who benefit from healthy diets and physical activity [122]. A survey carried out in Israel on caregivers of pediatric patients with chronic respiratory disorders, including BPD, showed that during the first-wave lockdown, clinical status worsened in 10% of these patients [122]. In addition, staying at home for long periods causes a reduction in vitamin D production, which may increase the risk of infections in children with chronic lung disease [123]. All these aspects need to be taken into account when caring for infants with BPD in these particular times.

## 8. Conclusions

Classifying and defining the endotypes of extreme prematurity and the phenotypes of BPD requires the close collaboration of obstetricians, neonatologists, pulmonologists, cardiologists, radiologists, pathologists, and epidemiologists. All this collaborative knowledge should result in a tailored approach to BPD diagnosis and therapy. In order to achieve this objective, it is also necessary to develop machine learning algorithms and artificial intelligence technologies that are capable of bringing together clinical, functional, imaging, and biomarker data.

## Figures and Tables

**Figure 1 jpm-12-00687-f001:**
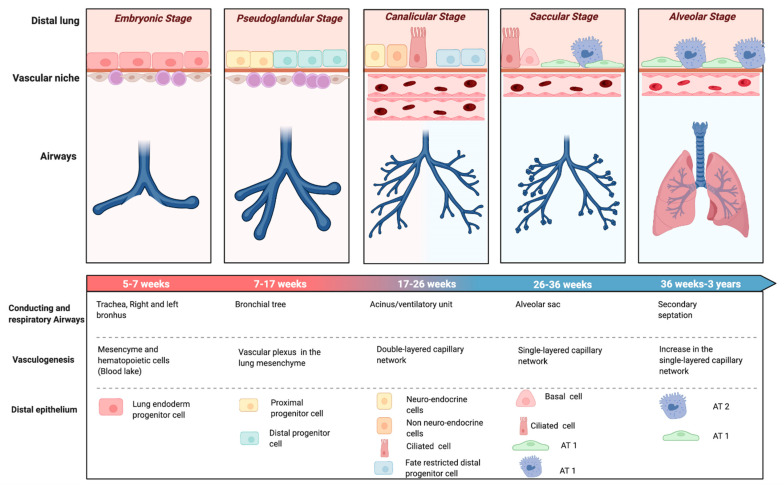
Stages of lung development. Evolution of the airways, epithelial and vascular components of the distal lung through pulmonary development. Abbreviations: Alveolar type 1 cells (AT1), Alveolar type 2 cells (AT2). Created with BioRender.com.

**Figure 2 jpm-12-00687-f002:**
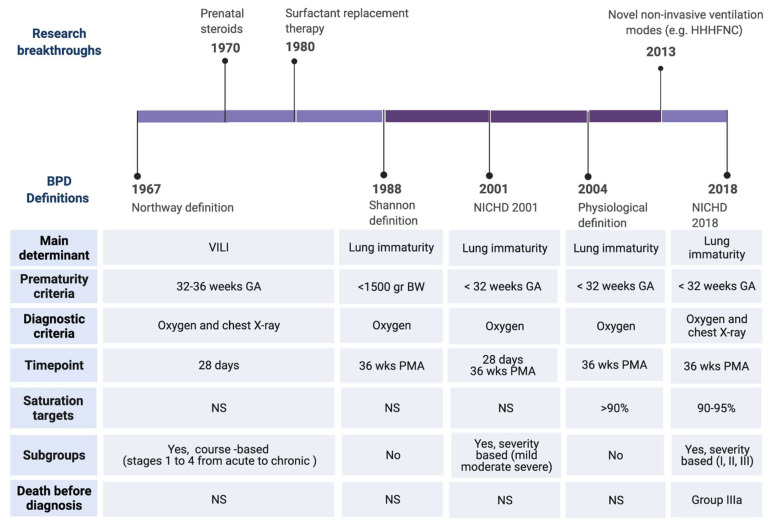
Timeline of BPD definitions. Upper: Major clinical breakthrough that have influenced the change of BPD definitions. Lower: Major differences among the definitions. Abbreviations: Heated, humidified, high flow nasal cannula (HHHNC), National Institute of Child Health and Human Development (NICHD), ventilator induced lung injury (VILI), birth weight (BW), gestational age (GA), post-menstrual age (PMA). Created with BioRender.com, [10,11,11,11,11,11,11,11].

**Figure 3 jpm-12-00687-f003:**
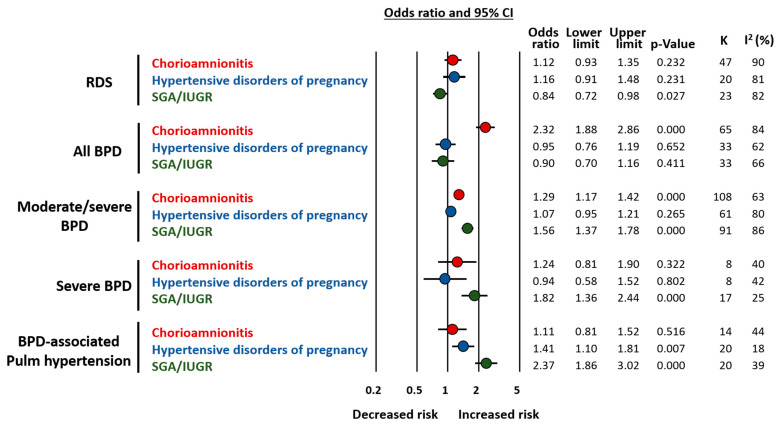
Summary of meta-analyses on the association between endotype of prematurity and short-term respiratory complications. Based on References [47,48]. Abbreviations: Respiratory distress syndrome (RDS), bronchopulmonary dysplasia (BPD), small for gestational age (SGA), intrauterine growth restriction (IUGR), confidence interval (CI).

**Figure 4 jpm-12-00687-f004:**
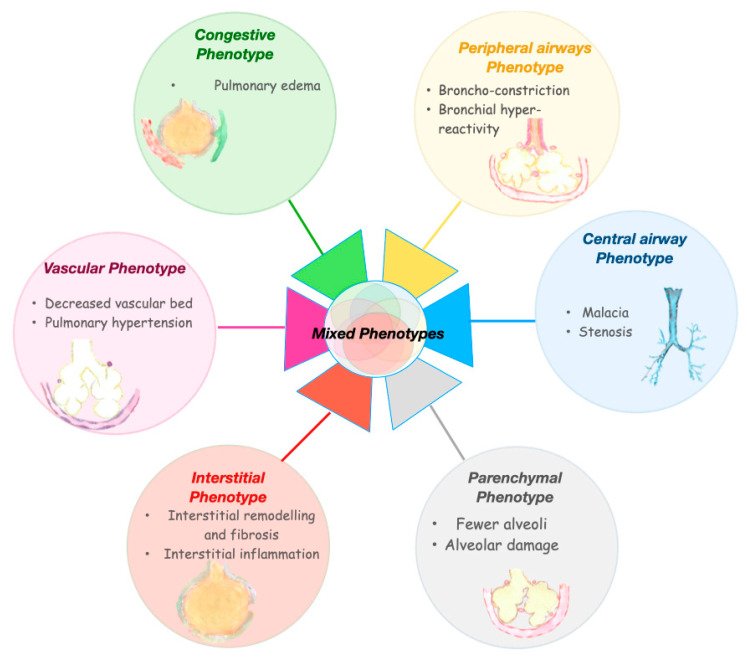
Bronchopulmonary dysplasia phenotypes.

**Figure 5 jpm-12-00687-f005:**
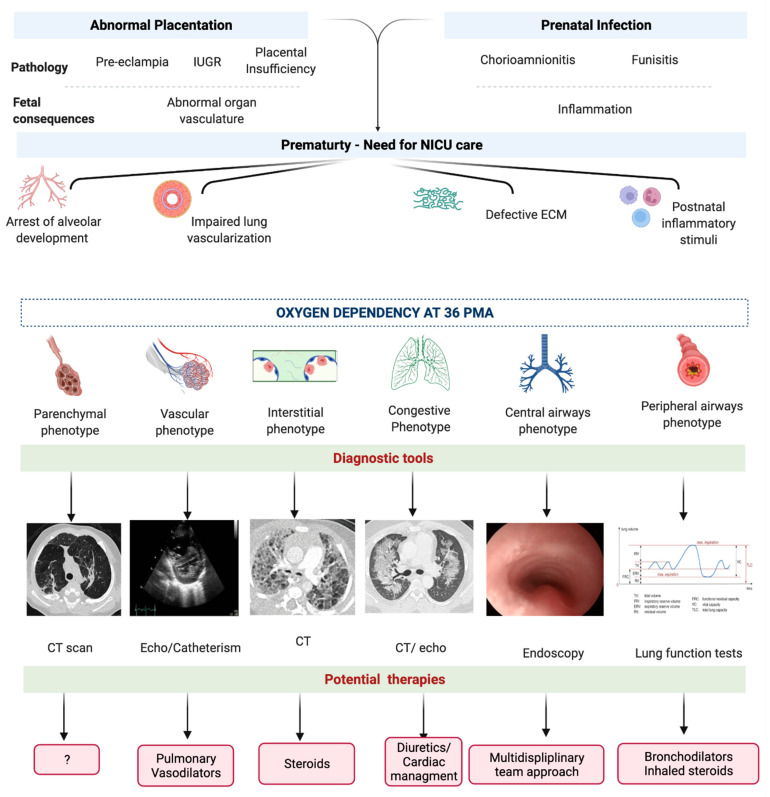
Diagnostic and therapeutic perspectives on phenotypes of bronchopulmonary dysplasia. *Abbreviations*: intrauterine growth restriction (IUGR), extra-cellular matrix (ECM), computed tomography (CT). Created with BioRender.com.

## Data Availability

Not applicable.

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
