# Peer review of "Endotypes of Prematurity and Phenotypes of Bronchopulmonary Dysplasia: Toward Personalized Neonatology"

_jpm, 2022, doi:10.3390/jpm12050687_

Round 1
Reviewer 1 Report
The review related with Bronchopulmonary dysplasia -- the disease a pathological reparative response of the developing lung to both antenatal and postnatal injury. The phenotypes and pathogenesis of the disease described in details.
- Maybe some additional description of therapeutic approach used in the clinical practice especially take into account different phenotypes. Treatment efficacy composition across phenotypes also will be interesting.
- Currently there are about 100 active clinical trials for BPD and about 150 completed in last 10 years. May be it will be reasonable to find "popular" phenotypes in active trials and compare treatment efficacy of "popular" and "rare" studies phenotypes.
- Take into account current epidemiological situation and especially burden of post-COVID respiratory sequel, it will be very important to add couple of sentence about COVID and post-COVID influence on Bronchopulmonary dysplasia development.
Author Response
1. Maybe some additional description of therapeutic approach used in the clinical practice especially take into account different phenotypes. Treatment efficacy composition across phenotypes also will be interesting.
Thank you very much for your constructive comments and observations. Following your recommendation, we have restructured section 7 (lines 442 to 548) to give more emphasis to the aspects of personalized medicine. We have also concentrated in this chapter the information on therapeutic strategies related to the different phenotypes of BPD (lines 471 to 501).
2. Currently there are about 100 active clinical trials for BPD and about 150 completed in last 10 years. May be it will be reasonable to find "popular" phenotypes in active trials and compare treatment efficacy of "popular" and "rare" studies phenotypes.
Following your suggestion, we have reviewed the randomized controlled trials on BPD registered at www.clinicaltrials.gov. As noted in the new version of the manuscript, most of these studies do not take into account BPD phenotypes. This new information can be found in lines 512 to 517 of the new version of the manuscript.
3. Take into account current epidemiological situation and especially burden of post-COVID respiratory sequel, it will be very important to add couple of sentence about COVID and post-COVID influence on Bronchopulmonary dysplasia development.
Following your suggestion, we have included two paragraphs (lines 528 to 548) on the potential consequences of the COVID-19 pandemic for BPD.
Reviewer 2 Report
I read with interest a review article by Pierro et al. on the endotypes of prematurity and its impact on the fetal lung development, leading to the formation of BPD. There is still knowledge gap in this area which requires more studies.
Even though the title of the manuscript is about personalized medicine, regrettably in this manuscript, there is no emphasis on this aspect of the disease. How the treatment can be catered to each endotypes of prematurity and different phenotypes of BPD should be discussed in more details. For instance, to include lists of relevant clinical trials or animal models in BPD in regard to personalized/ precision medicine. Section 7.2 should be expanded further.
Also suggest adding in a section on future prospect focusing on future research focus in the hope to improve survival of these unfortunate group of neonates, towards personalized medicine.
Author Response
Even though the title of the manuscript is about personalized medicine, regrettably in this manuscript, there is no emphasis on this aspect of the disease. How the treatment can be catered to each endotypes of prematurity and different phenotypes of BPD should be discussed in more details. For instance, to include lists of relevant clinical trials or animal models in BPD in regard to personalized/ precision medicine. Section 7.2 should be expanded further.
Also suggest adding in a section on future prospect focusing on future research focus in the hope to improve survival of these unfortunate group of neonates, towards personalized medicine.
Thank you very much for your constructive comments and suggestions. Following your suggestions, we have restructured Section 7(lines 442 to 548). In the new version of the manuscript, more emphasis is placed on personalized medicine both at the level of diagnostic and therapeutic strategies and at the level of future research.